# Steady-state measures of visual suppression

**Daniel H. Baker**[1]*, **Greta Vilidaite**[2], **Alex R. Wade**[1]

**1** Department of Psychology and York Biomedical Research Institute, University of York, York, United Kingdom, **2** School of Psychology, University of Southampton, Southampton, United Kingdom

* daniel.baker@york.ac.uk

**Data Availability Statement:** All data and scripts are publicly available at: https://dx.doi.org/10.17605/OSF.IO/E62WU.

**Funding:** DHB supported by grant number RG130121 from the Royal Society (https://royalsociety.org/). The funders had no role in study

## Abstract

In the early visual system, suppression occurs between neurons representing different stimulus properties. This includes features such as orientation (cross-orientation suppression), eye-of-origin (interocular suppression) and spatial location (surround suppression), which are thought to involve distinct anatomical pathways. We asked if these separate routes to suppression can be differentiated by their pattern of gain control on the contrast response function measured in human participants using steady-state electroencephalography. Changes in contrast gain shift the contrast response function laterally, whereas changes in response gain scale the function vertically. We used a Bayesian hierarchical model to summarise the evidence for each type of gain control. A computational meta-analysis of 16 previous studies found the most evidence for contrast gain effects with overlaid masks, but no clear evidence favouring either response gain or contrast gain for other mask types. We then conducted two new experiments, comparing suppression from four mask types (monocular and dichoptic overlay masks, and aligned and orthogonal surround masks) on responses to sine wave grating patches flickering at 5Hz. At the occipital pole, there was strong evidence for contrast gain effects in all four mask types at the first harmonic frequency (5Hz). Suppression generally became stronger at more lateral electrode sites, but there was little evidence of response gain effects. At the second harmonic frequency (10Hz) suppression was stronger overall, and involved both contrast and response gain effects. Although suppression from different mask types involves distinct anatomical pathways, gain control processes appear to serve a common purpose, which we suggest might be to suppress less reliable inputs.

## Author summary

Suppression is a ubiquitous process in the brain that acts to regulate neural activity. In the visual system, there are multiple suppressive pathways, which normalize neural responses across dimensions such as spatial location, eye of origin, and feature (e.g. stimulus orientation). In this study we ask how we can characterise suppression from these different sources by examining the effect on the contrast response function (the function mapping the physical stimulus intensity to the neural response), measured using steady-state electroencephalography (EEG). Using a Bayesian computational model, we estimate the likely range of model parameter values summarising contributions from contrast gain

design, data collection and analysis, decision to publish, or preparation of the manuscript.

**Competing interests:** The authors have declared that no competing interests exist.

(which shifts the contrast response function laterally) and response gain (which scales it vertically). A meta analysis of 16 previous studies provided inconclusive evidence, but generally supported a greater contribution from contrast gain control. Data from two new studies showed convincing effects of contrast gain control for four different mask types at the first harmonic (stimulus flicker frequency), and both contrast and response gain at the second harmonic (twice the flicker frequency). We discuss these findings in the context of work using other experimental methodologies, and speculate about the purpose of suppression in the nervous system.

## 1 Introduction

Suppression is a fundamental component of the nervous system, and is critically important for modulating neural firing [1]. Without suppression, neural activity would be too metabolically expensive, and uncontrolled excitation might lead to seizures. In the visual system, neurons responsive to a spatially localised narrowband target stimulus are suppressed by nearby neurons that differ in their tuning [2]. This tuning can involve different orientations (cross-orientation suppression), different spatial locations (lateral, or surround suppression), and different eye-of-origin (interocular suppression). Suppression is typically studied using a masking paradigm, where the response to a target stimulus is reduced by the presence of a high contrast mask (see examples in Fig 1A).

Several studies have demonstrated that these different types of suppression have distinct characteristics, and may occur at different stages in the early visual pathway. For example, suppression from an overlaid mask shown to the same eye as a target is immune to adaptation [3, 4], occurs at temporal frequencies above the range at which cortical neurons respond [4–7], and therefore appears consistent with a pre-cortical locus [4, 6]. If a mask is presented dichoptically (to the opposite eye from the target), suppression can be reduced by adaptation [3, 6, 7], has a temporal profile consistent with cortical neurons [6, 7], and is reduced by applying biculline (a compound that blocks the suppressive neurotransmitter GABA) to early visual cortex [7]. This points to a cortical locus for interocular suppression. Finally, surround masks have tighter tuning than overlaid masks and are most effective in the periphery [8], can be adapted [9], and (in V1) cause suppression via feedback from higher visual areas [10]. Additionally, some studies have linked the magnitude of surround suppression with endogenous levels of GABA in early visual cortex [11, 12], again pointing to a cortical locus.

An important distinction concerns whether a suppressive effect modulates the contrast gain or the response gain of a neuron (or neural population). Changes in contrast gain shift the stimulus-response curve (the contrast response function) laterally, whereas changes in response gain scale the function vertically (see examples in Fig 1C). These different patterns may be indicative of specific neurophysiological underpinnings for an effect, and potentially different processes might occur at successive stages of processing. Sengpiel et al. [13] showed that in V1, dichoptic and surround masks primarily affected response gain, whereas overlaid masks affected contrast gain. Other studies have found that spatial attention modulates response gain [14], whereas suppression from overlaid masks is more consistent with contrast gain [15]. Spatial adaptation appears to affect both contrast and response gain in primary visual cortex [16], whereas motion adaptation is mostly attributable to contrast gain in area MT [17]. Work by Hou et al. [18] used source-imaged EEG to study interocular suppression, and found evidence of a transition from contrast gain effects in V1 to response gain effects in extra-striate areas. In addition, there is evidence that suppression builds up at successive stages

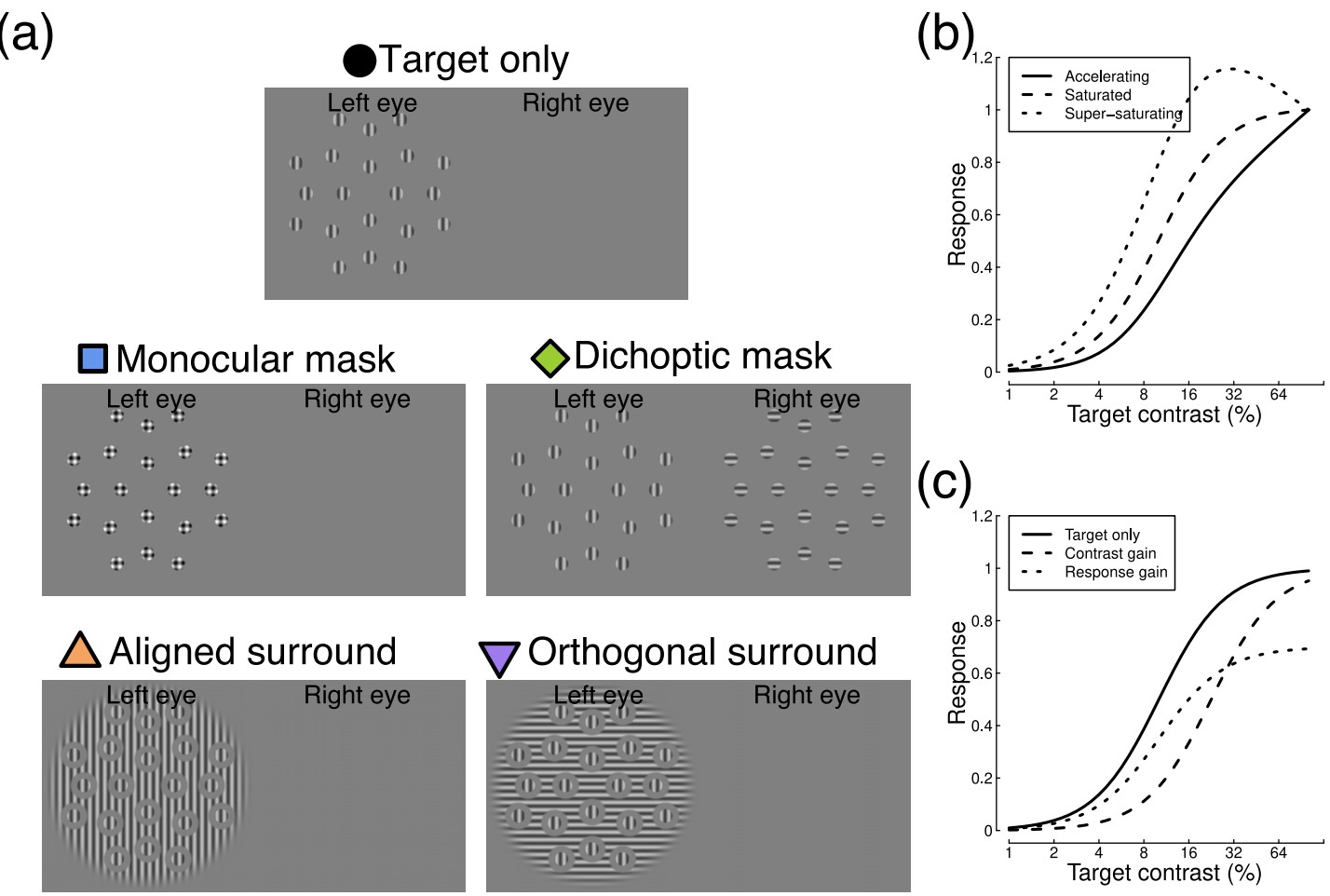

**Fig 1. Example stimuli and illustration of contrast response functions.** Panel A shows five stimulus arrangements, illustrating how a vertical target pattern can be combined with four different mask types. Panel B shows three varieties of contrast response function, that either continue to accelerate (solid line), saturate (dashed line) or super-saturate (dotted line) across the range of displayable stimulus contrasts. Panel C illustrates a contrast gain (dashed line) and a response gain (dotted line) shift, relative to a baseline response (solid line).

of cortical processing beyond primary visual cortex, and is stronger at later levels in the visual hierarchy [19]. This is especially likely for surround suppression, which might be mediated by higher-level neurons with large receptive fields.

Neural responses can be measured non-invasively using steady-state visual evoked potentials (SSVEPs; [20]) typically recorded in humans using either electroencephalography (EEG) or magnetoencephalography (MEG). By flickering the target stimulus at a fixed frequency, entrained neural oscillations are evoked at the flicker frequency, and also its higher harmonics (integer multiples of the flicker rate). Previous studies have shown that contrast-response functions measured using SSVEP are strongly modulated by overlaid masks [15, 21, 22], dichoptic masks [23, 24], and surround masks [25–27].

We begin by conducting a computational re-analysis of 16 published studies to determine whether each type of suppression is best characterized as a contrast gain or a response gain effect. We then report results from two new SSVEP experiments to directly compare four mask types using a common protocol. This also allowed us to explore changes across different electrode sites and different response frequencies. A secondary aim was to determine whether

suppressive signals saturate as a function of contrast. We conducted the main experiment with two different mask contrasts, and analyse the data using a hierarchical Bayesian modelling approach.

## 2 Methods

### 2.1 Ethics statement

The study was approved by the Department of Psychology Ethics Committee at the University of York (identification number 113). All participants provided written informed consent prior to testing.

### 2.2 Computational meta-analysis

**2.2.1 Inclusion criteria.**   Studies were included if they reported steady-state contrast response functions measured in human adults with no known disorders or medical conditions. Responses at 3 or more target contrasts were required to fit the baseline functions. We also required that a mask stimulus was presented in at least one condition. This could either be overlaid, dichoptic (presented to the opposite eye from a monocular target), or surrounding the target. We excluded one study with flanking masks which reported only facilitation [28]. We divided the surround conditions into those where the surround was aligned with (parallel to) the target, and those where it was orthogonal. For the overlay and dichoptic conditions, some studies used gratings and others used noise stimuli. Where multiple masking conditions were reported, we included data at the lowest mask contrast tested, and used data with orthogonal masks in preference to aligned masks (for overlay and dichoptic conditions). In studies where an experimental manipulation was carried out, we used data from the baseline (pre-manipulation) condition. We searched online databases using search terms including *SSVEP*, *steady-state*, *dichoptic*, *surround*, *mask* and *suppression*, and applied the above criteria, resulting in 16 studies for inclusion in the analysis.

**2.2.2 Analysis and modelling.**   Contrast response function data were extracted using a computer program (WebPlotDigitizer, [29]) from the figures in each paper. Where necessary, these were converted to signal-to-noise ratios by dividing by the response to a blank screen, or at adjacent frequency bins to the target, or to the lowest target contrast condition. In one case [30], results were averaged across different temporal frequency conditions to provide a single data set for the study, but we confirmed that this did not affect our conclusions by repeating the analysis for each temporal frequency separately.

Our primary objective was to understand the relative contributions of contrast gain and response gain to suppression from different mask types. We quantified this using a two-stage modelling approach. At the first stage, we fitted a standard gain control model [2] with three free parameters to the baseline data using a downhill simplex algorithm. The model is defined as:

$$resp = R_{max} \frac{C^p}{Z + C^2} + 1, \qquad (1)$$

where *C* is the target contrast. The *Z* parameter sets the horizontal position of the response curve, *p* governs the function shape (see Fig 1B) from accelerating ($p > 2$) to saturating ($p = 2$) to super-saturating ($p < 2$), and $R_{max}$ scales the overall height of the function. The additive constant (+1) represents additive noise, and converts the model response to a signal-to-noise ratio (implicitly, we also divide *resp* by 1, but this is omitted as it has no effect). We fitted the model independently to each study's baseline data by minimising the root-mean-squared error between model and data.

The second stage of fitting used the parameter estimates from the first stage, and fitted the responses in the presence of a mask using the equation:

$$resp = \frac{R_{max}}{r} \times \frac{C^p}{gZ + C^2} + 1, \tag{2}$$

where the new terms $r$ and $g$ are free parameters that govern the extent of response gain and contrast gain, respectively. Values of $r, g > 1$ indicate suppression, though in principle masks can also cause facilitation ($r, g < 1$). Note that even with the value of $p$ fixed by the first stage of modelling, the $r$ and $g$ parameters can accommodate apparent changes in the steepness of the contrast response function and the extent of saturation, by moving the steepest part of the curve to higher contrasts ($g$), or by reducing the available range of responses ($r$). Our rationale for fitting the baseline data first is that the values of the $p$, $Z$ and $R_{max}$ parameters are then fixed, and cannot trade off with the suppression-specific parameters ($r$ and $g$) when fitting the masking data. This gives a cleaner estimate of the relative effects of each type of suppression.

We estimated values of the suppression parameters jointly using the data from all studies (separately for each mask type) in a hierarchical Bayesian model. We defined broad hyper-priors for $g$ and $r$ as gamma distributions, with parameters $\alpha = 1.5$, $\beta = 0.5$. These functions peak at $\frac{\alpha-1}{\beta} = 1$, so the modal assumption before observing any data is that there is no suppression of either kind. The priors had greater probability mass at values $> 1$, reflecting our expectation that one or both parameters would produce suppression, but also extended below 1, ensuring that the model was capable of capturing facilitation where it appeared in the data. Both parameters were constrained to have positive values. Bayesian modelling was implemented in *Stan* [31], based on an example script for hierarchical nonlinear regression accompanying Chapter 17 of ref [32]. We examined how the posterior distribution of each parameter varied with mask type, both for individual studies, and across the whole sample.

## 2.3 EEG experiments

**2.3.1 Participants.**   Twelve participants completed each version of the experiment; 3 participants completed both experiments, the remaining 9 were unique to each experiment. All participants had normal or corrected-to-normal vision, and no known visual abnormalities. Participants were briefed on the experimental protocols and purpose, and provided written informed consent.

**2.3.2 Apparatus and stimuli.**   Stimuli were presented using a ViewPixx 3D display (VPixx Technologies Inc., Quebec, Canada), driven by a Mac Pro computer. The refresh rate was 120 Hz, and we interleaved frames intended for the left and right eyes (60 Hz refresh rate per eye). To enable stereo presentation, the display update was synchronised with a set of NVidia 3D pro active shutter glasses using an infra-red signal. The display had a resolution of 1920 × 1280 pixels, and was viewed from a distance of 57cm, at which one degree of visual angle subtended 36 pixels. To ensure good contrast resolution, the display was run in the high bit-depth monochrome M16 mode, which provided 16 bits of greyscale resolution. A Minolta LS110 photometer was used to gamma correct the display, which had a maximum luminance of 102 $cd/m^2$.

All stimuli were patches of sinusoidal grating with a spatial frequency of 1 cycle per degree. Target stimuli were randomly oriented on each trial, and windowed by a raised cosine envelope with a width of 2 degrees. There were 20 targets arranged in a symmetrical pattern around a central fixation marker, as shown in Fig 1A. The target eccentricities were 3.6, 7.1, 8.5 and 10.7 degrees from the central fixation. Stimuli were spaced in 90 degree intervals at each radius, or in 45 degree intervals at the largest eccentricity. All target stimuli flickered sinusoidally at 5Hz (on-off flicker), between 0% contrast and their nominal Michelson contrast,

which was one of six values (0, 6, 12, 24, 48 and 96%). Percentage Michelson contrast is defined as $100\frac{L_{max}-L_{min}}{L_{max}+L_{min}}$, where $L$ is luminance. Targets were shown to one eye only, which was chosen randomly on each trial. A binocular fixation marker was created from a cluster of overlaid squares (each 13 arc min wide) with random grey levels, and shown to both eyes to aid binocular fusion. Similar markers were also presented in the four corners of the stimulus region, at a distance of 15.7 degrees from the display centre.

We measured target responses with no mask, and also with four categories of mask stimulus. Monocular masks were shown to the same eye as the targets and in the same locations, but had orthogonal orientation. Dichoptic masks were the same, but shown to the non-target eye. Aligned surround masks were large (28 degrees in diameter) grating patches with the same orientation as the target, and with holes surrounding each target element (and the fixation marker). The holes were 4 degrees in diameter, meaning the gap between target and mask was 1 degree (one cycle of the stimulus waveform). Orthogonal surrounds were the same, but were oriented at 90 degrees relative to the targets. Both surround masks were presented to the target eye. There were two principal mask contrasts that were used in the two versions of the experiment: 12% and 24%. We also tested several additional mask contrasts (6, 48 and 96% contrast) at a single target contrast of 24%. The masks drifted at a speed of 6 deg/sec so that the phase alignment between mask and target changed over time [27]. Note that drifting gratings do not produce a steady-state signal, so we did not record responses to the mask stimuli. In addition, for some of the monocular mask conditions, the highest target contrast was reduced from 96% to 88% or 68% contrast to avoid clipping artifacts caused by overlaying the target and mask.

EEG activity was recorded using a 64-channel ANT Neuroscan system sampling at 1 kHz. Participants wore Waveguard caps, with electrodes organised according to the 10/20 system. The ground was located at position *AFz*, and each channel was referenced to the whole-head average. Electrode impedance was maintained at or below 5 kΩ throughout the experiment. Digital parallel triggers were sent from the ViewPixx display to the EEG amplifier, and recorded the onset of each trial on the EEG trace. Data were amplified, digitised, and saved to disc for offline analysis.

**2.3.3 Procedure.**   After providing consent, participants were set up with an EEG cap of appropriate size. They then completed six blocks, each comprising a full repetition of the experiment. Blocks lasted around 10 minutes, with the opportunity to take breaks between blocks. Within each block, all 42 conditions were repeated once in a randomized order. Trials lasted 11 seconds, with an inter-trial interval of 3 seconds. Participants were asked to monitor the central fixation and, as far as possible, to minimise blinking when a stimulus was displayed. To maintain attention, the central fixation marker was changed occasionally by re-randomizing the positions and luminances of the squares. There was a 50% chance of this happening on each trial. Participants were asked to count the number of times the fixation marker changed, and report this at the end of the block.

**2.3.4 Data analysis and modelling.**   All data were converted from the native ANT-EEP-robe format to a compressed comma-separated value (csv) text file using a custom *Matlab* script and components of the EEGlab toolbox [33]. The data for each participant were then loaded into *R* for analysis. A ten-second waveform for each trial at each electrode was extracted, omitting the first one second after stimulus onset to avoid transients. The fast Fourier transform was calculated for each waveform, and the spectrum stored in a matrix. All repetitions of each condition were then coherently averaged (i.e. averaging the full complex Fourier spectrum, including both phase and amplitude components), before being converted to a signal-to-noise ratio by dividing the amplitude at each frequency by the mean amplitude of the neighbouring 10 bins (±0.5 Hz in steps of 0.1 Hz). The signal-to-noise ratio at the target

flicker frequency (5 Hz) and its second harmonic (10 Hz) were then used as dependent variables for further analysis.

We modelled the data using a two-stage Bayesian hierarchical model similar to that described above for the computational meta-analysis. Here, participant was the unit of observation instead of study. The other main difference was that we also used a hierarchical model (instead of simplex fitting) at the first stage to fit the parameters of the baseline contrast response function ($Z$, $p$ and $R_{max}$). This seemed appropriate for our novel data set, given that all participants viewed the same stimuli, whereas in the computational meta-analysis different studies had different stimulus parameters. The hyperpriors for each parameter were normal distributions with parameters: $\mu = 100$, $\sigma = 40$ ($Z$); $\mu = 2$, $\sigma = 0.25$ ($p$); and $\mu = 5$, $\sigma = 2$ ($R_{max}$). All parameters were constrained to have positive values. Again, we were most interested in the posterior distributions of the suppression parameters ($r$ and $g$), and explored how these varied by mask type, electrode position, and response frequency.

## 3 Results

### 3.1 Previous studies do not sufficiently distinguish contrast vs response gain

We began by conducting a computational meta-analysis of 16 SSVEP studies from the literature [15, 18, 21–26, 30, 34–40]. Study-specific information is given in Table 1 and the results are shown in Fig 2. For each study, we replot the contrast response functions for the target alone (black points), and with the mask present (coloured points), along with model fits (curves). The model described the data well. The kernel density functions show posterior distributions of parameter estimates for the response gain parameter ($r$, grey distributions), and the contrast gain parameter ($g$, coloured distributions). For each mask type, the vertical dashed line indicates no suppression (a weight of 1). The 95% highest density intervals are given by the horizontal bars—where these overlap 1 we lack credible evidence for that type of suppression.

For individual studies, we see credible evidence for both contrast gain (9 data sets) and response gain (4 data sets). This is most consistent for the overlay masks, which are generally well explained by contrast gain control. However, the 95% highest density interval of the group posterior distribution for contrast gain control, shown in the final row, overlapped with a value of 1. This indicates that we do not have credible evidence for a contrast gain effect for overlay suppression. The other three mask types had a similar outcome, as the 95% highest density intervals of the group posterior distributions for both parameters all overlap 1. This suggests that overall the literature does not give a consistent picture of whether response gain or contrast gain is responsible for different types of suppression (though the parameter values for contrast gain are somewhat higher on average). This could be for any number of reasons, but is likely to be partly due to the methodological heterogeneity across studies (see Table 1). To address this shortcoming, we conducted a new study in which participants viewed stimuli involving all four types of mask.

### 3.2 Suppression is due to contrast gain at the first harmonic for all mask types

In our empirical experiments, the target stimulus evoked strong steady-state responses at both the first harmonic frequency (5 Hz) and the second harmonic frequency (10 Hz). Fig 3A shows the averaged Fourier spectrum from the baseline (no mask) condition with 96% target contrast. Responses at both frequencies were strongest at the occipital pole, over early visual

**Table 1. Table summarising methodological details for each study in the meta analysis.** N: number of participants, SI: saturation index.

| Source | Method | N | Target | TF (Hz) | Mask | Location | SI |
|---|---|---|---|---|---|---|---|
| Baker (2014) [21] Fig 3A | EEG | 6 | 1 c/deg grating | 7 | Orthogonal overlay | Oz, POz | 0.22 |
| Burr (1987) [35] Fig 4A | EEG | 1 | 2 c/deg grating | 7.8 | Orthogonal overlay | Oz | -0.44 |
| Busse (2009) [22] Fig 6C | EEG | 5 | 1 c/deg grating | 4.5 | Orthogonal overlay | V1 (source localised) | 0.09 |
| Candy (2001) [30] Fig 3 | EEG | 8 | 1 c/deg grating | 3.3, 5.5 | Orthogonal overlay | Oz, O1, O2 | 0.31 |
| Pei (2017) [36] Fig 4 | EEG | 10 | Binary noise | 5.14 | Noise overlay | Oz | -0.03 |
| Ross (1991) [37] Fig 1 | EEG | 1 | 2 c/deg grating | 8.8 | Orthogonal overlay | Oz | -0.04 |
| Smith (2017) [38] Fig 2A | EEG | 28 | 0.5 c/deg grating | 7 | Orthogonal overlay | Oz, POz, O1, O2 | 0.35 |
| Tsai (2012) [15] Fig 3A | EEG | 10 | Binary noise | 5.14 | Noise overlay | V1 (source localised) | -0.13 |
| Baker (2015) [34] Fig 2A | EEG | 5 | Binary noise | 10 | Dichoptic | Oz, POz | 0.27 |
| Baker (2017) [23] Fig 4D | EEG | 12 | 1 c/deg grating | 5 | Dichoptic | Oz, POz | 0.04 |
| Chadnova (2018) [24] Fig 2 | MEG | 5 | Binary noise | 4 | Dichoptic | V1 (source localised) | 0.33 |
| Hou (2020) [18] Fig 4A | EEG | 15 | 2 c/deg grating | 8.5 | Dichoptic | V1 (source localised) | 0.09 |
| Zhou (2015) [40] Fig 2 | EEG | 12 | Binary noise | 10 | Dichoptic | Oz | 0.22 |
| Benjamin (2018) [25] Fig 6 | EEG | 13 | 2 c/deg grating | 7 | Aligned and Orthogonal Surround | Oz | -0.01 |
| Vanegas (2015) [26] Fig 3 | EEG | 21 | 1 c/deg grating | 25 | Aligned and Orthogonal Surround | POz | 0.38 |
| Vanegas (2019) [39] Fig 3B | EEG | 11 | 1 c/deg grating | 25 | Aligned Surround | Pz, POz, Oz | 0.49 |

cortex (see inset scalp plots). At most electrodes, responses increased monotonically as a function of contrast (see examples in Fig 3B and 3C). In general, responses at the first harmonic (5Hz) were more likely to accelerate, and those at the second harmonic more likely to saturate or super-saturate. The scalp plot insets to Fig 3B and 3C summarise this using a saturation index proposed by Ledgeway et al. [41]. It is calculated by taking the difference between the responses at the highest two contrasts (96% and 48%), and dividing by the maximum response. Positive values of SI correspond to acceleration (plotted violet), SI = 0 to saturation (white), and negative SI values to super-saturation (green). Notice that overall the first harmonic responses accelerate (median SI = 0.10), but that many of the second harmonic responses saturate or super-saturate (median SI = 0.01).

To quantify how suppression varied across the scalp, and across different mask types and response frequencies, we fitted a hierarchical Bayesian model to the data. The first stage of this process involved estimating values for the three free parameters in Eq (1). Fig 4A shows an example fit at electrode *Oz* for the low contrast mask experiment. The thick black curve is the fit using the mean posterior parameter estimates ($p = 1.93$, $Z = 134.7$, $R_{max} = 4.83$), and thin lines show predictions for 100 randomly sampled posterior parameter combinations. At the second stage of fitting, we estimated values of the suppressive parameters *g* and *r* for each mask type. Example fits are shown in Fig 4B–4E, with accompanying posterior distributions of parameter estimates in panels G-J. Note that the parameters are estimated individually for each participant, and the plots in Fig 4 show group level parameters, which do not necessarily correspond to the average data as well as an optimal least-squares fit (the $R^2$ values given in each plot likewise correspond to the fit to the individual data). We assess whether a parameter makes a credible contribution to the response by determining whether the 95% highest density interval of the posterior (shown by the black bars at the margins of Fig 4G–4J) exceeds 1. For all four examples shown in Fig 4G–4J, the contrast gain parameter (*g*, y-axis) was credibly greater than 1, whereas the response gain parameter (*r*, x-axis) was not credibly different from 1. This is evidence that all four mask types modulate responses via contrast gain control at electrode *Oz*, for the first harmonic response.

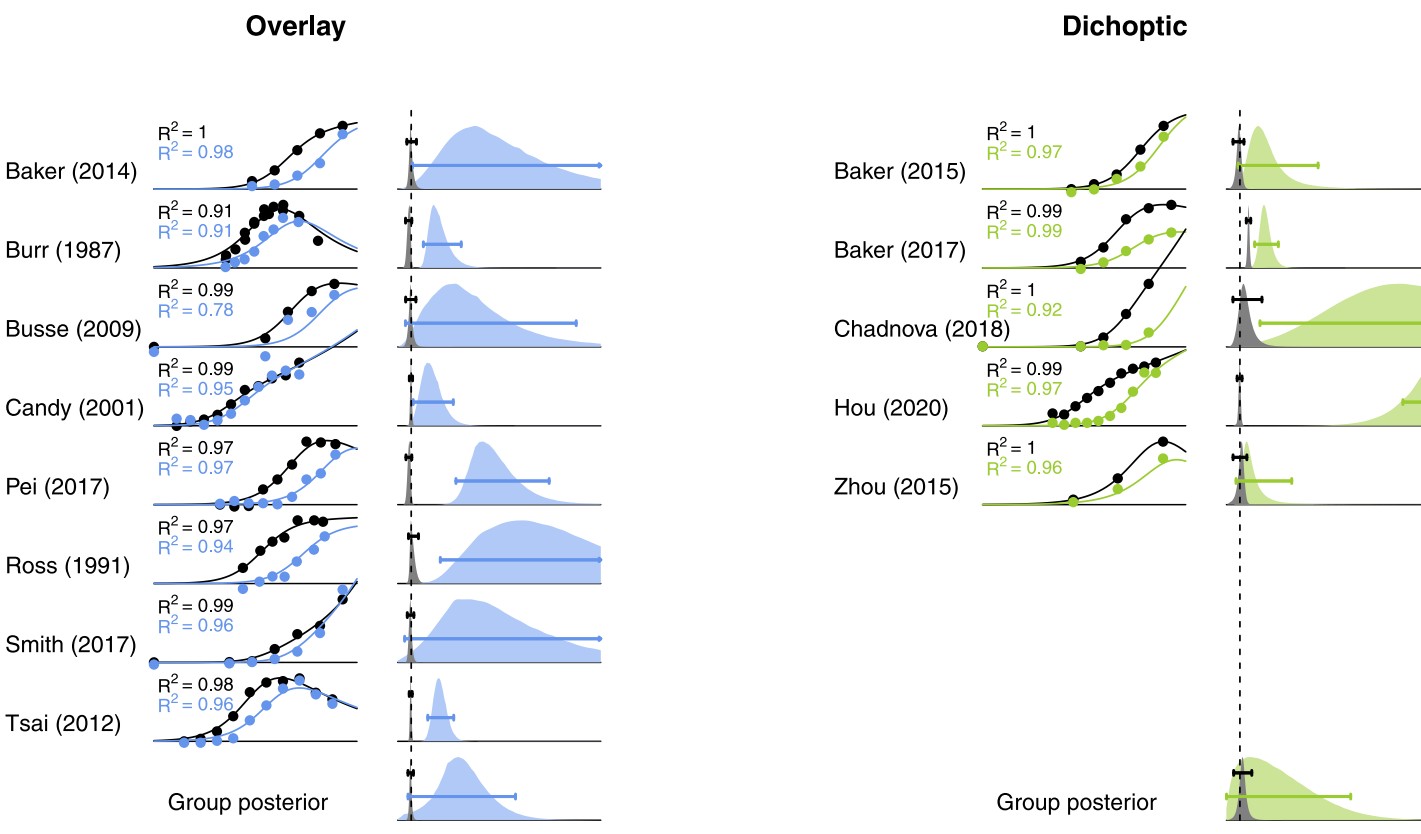

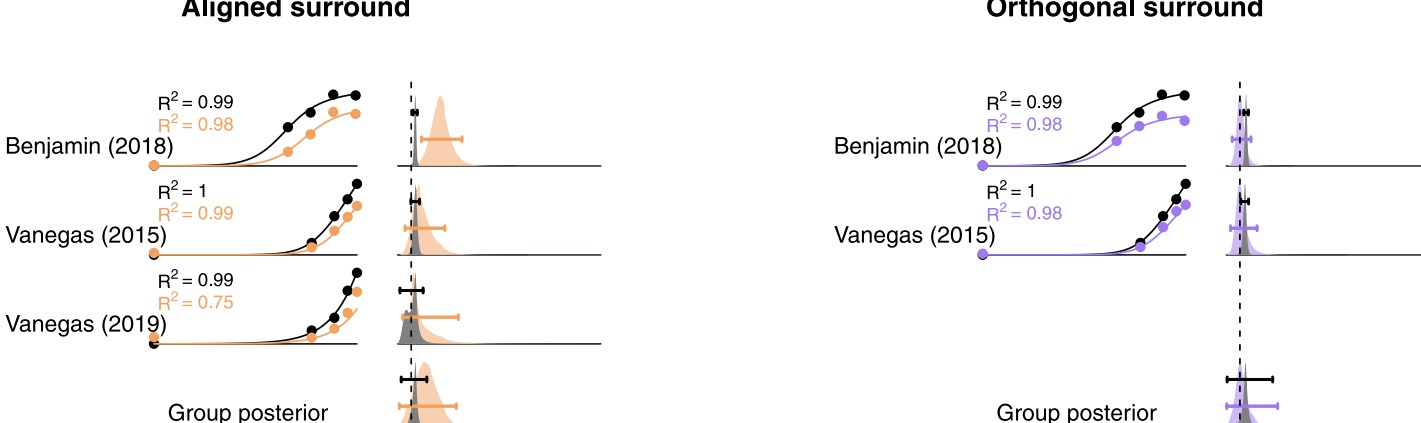

**Fig 2. Computational meta-analysis of 16 studies from the literature reporting SSVEP measures of suppression.** Each study is referred to by the first author surname —see text for full citations. Contrast response functions at baseline (black points) and with a mask present (coloured points) were fit using a two stage modelling procedure (curves). The posterior distributions (vertically rescaled for visibility) of parameter estimates for response gain (grey) and contrast gain (colours) are shown for each study and the group estimates. Vertical dashed lines indicate a parameter value of 1 (the axis extends to $x$ = 15). Horizontal bars give the 95 percent highest density intervals for each parameter estimate. $R^2$ values in each sub-panel indicate the coefficient of determination, calculated separately for the baseline (black text) and masked data (coloured text).

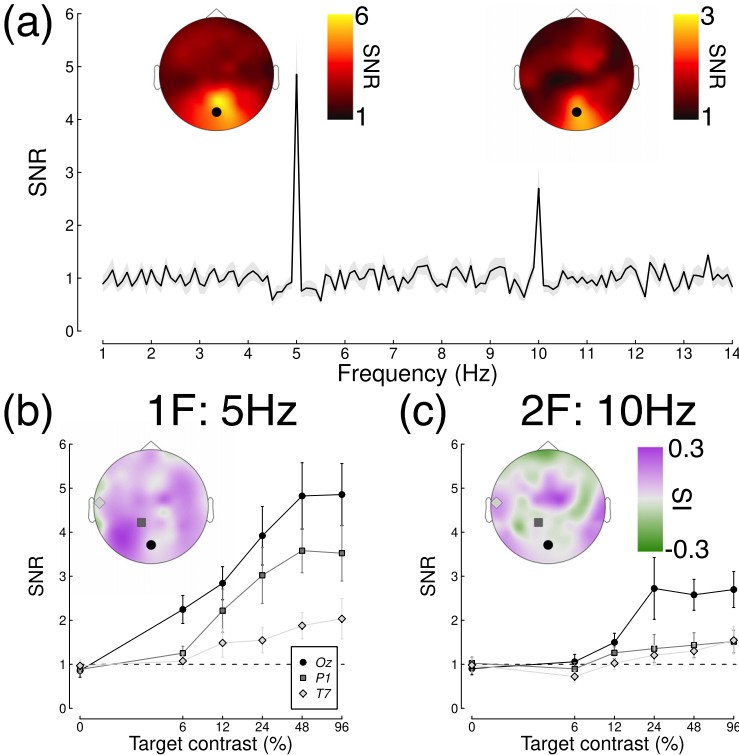

**Fig 3. Averaged Fourier spectrum and example contrast response functions.** Panel A shows the spectrum for a high contrast target, with inset scalp plots showing SNRs at the first and second harmonic frequencies. The spectrum is taken from electrode *Oz*, indicated by the black points in the scalp plots. The shaded region and error bars indicate ±1 standard error. Panels B and C show example contrast response functions at the first and second harmonics at electrodes *Oz*, *P1* and *T7*, averaged across participants (N = 12). The inset scalp plots show how the saturation index varies across the head.

## 3.3 Suppression across electrode and scalp location

We repeated the above analysis independently at each electrode, for each response frequency (5 Hz and 10 Hz), and for both experiments (12% and 24% mask contrast). Fig 5 summarises the results for the 12% mask contrast experiment, and for each mask type. For the first harmonic (5 Hz) response (top two rows), there were strong contrast gain control effects (panels A-D), but little credible effect of response gain (panels E-H). For the second harmonic response (10 Hz), although some contrast gain effects were credible at the occipital pole (electrode *Oz* for all mask types, panels I-L), suppression was also well described by response gain (panels M-P). Example contrast response functions and posterior distributions at the second harmonic are shown in Fig 6. This overall pattern was replicated in our second data set with higher (24%) contrast masks (Fig 7).

Closer inspection of these results reveals some interesting subtleties and differences across mask conditions. Note in particular that the contrast gain weights for surround suppression at the first harmonic are generally weaker at the occipital pole (electrodes *Oz* and *POz*) than for monocular and dichoptic suppression. We can use the posterior parameter distributions to calculate a Bayesian equivalent of a p-value for these comparisons to determine statistical significance. This analysis shows that at *Oz* less than 5% of the posterior distribution for aligned surrounds ($p = 0.006$) and orthogonal surrounds ($p = 0.003$) equals or exceeds the mean weight for monocular overlay masks (these are the distributions shown along the y-axes of Fig 4G, 4I

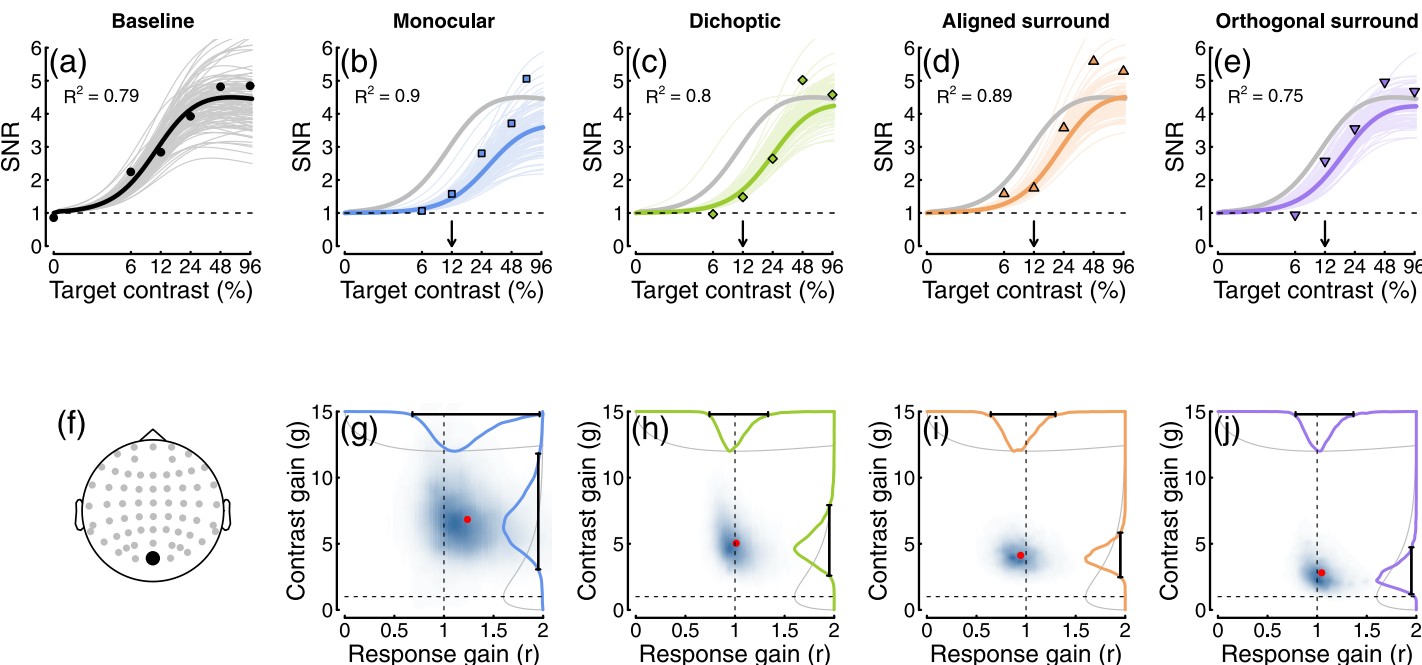

**Fig 4. Contrast response functions from electrode *Oz*, with example model fits and posterior parameter estimates.** Panel A shows the data from the baseline (no mask) condition (points), plotted alongside model curves for the mean posterior parameter estimates (thick curve), and random posterior samples (thin curves). Panels B-E show data for four types of mask in the same format (grey curves duplicate the mean fit from panel A), with the arrows indicating the mask contrast. The $R^2$ values are derived from the fits to the individual participant data, rather than the group averaged data plotted here. Panel F shows the electrode location. Panels G-J show posterior density estimates for the response gain (x-axis) and contrast gain (y-axis) weight parameters. Red points show the means, dashed lines give the value expected in the case of no effect (a weight of 1), and distributions along the margins show the prior (grey) and posterior (coloured) distributions for each parameter. For all mask types, the contrast gain weight estimate was substantially greater than 1.

and 4J). Suppressive weights also increase at bilateral electrodes over more parietal regions of cortex, particularly for surround suppression. For example, the weights are greater at electrode *P7* than *Oz* for both aligned ($p = 0.001$) and orthogonal ($p = 0.002$) surrounds. This might reflect increased suppression in extra-striate cortical regions that have larger receptive fields. More generally, it suggests that suppression builds up across successive stages of processing. It also appears that, whereas suppression at the first harmonic is primarily due to contrast gain control, suppression at the second harmonic involves changes in both contrast and response gain (see lower two rows of Figs 5 and 7). This may well reflect the involvement of different classes of neurons—for example, second harmonic responses imply more severe nonlinearities [42], which might include suppression. This is also consistent with the greater saturation of the second harmonic response (inset to Fig 3C).

### 3.4 Limited saturation of mask signals

Finally, we asked about the properties of the mask signal. Of particular interest is whether the mask signal itself saturates before suppressing the target. If it does, this implies the presence of a nonlinearity before suppression impacts, as has been shown psychophysically for surround masks [43]. Fig 8A shows model predictions for a linear suppressive signal (black curve), and a saturating suppressive signal (red curve). We therefore measured responses at a fixed (24%) target contrast, for mask contrasts that ranged from 0% to 96%. For this analysis, we pooled data across the two experiments, giving us N = 21 participants (data for the three participants who completed both experiments were averaged to give a single data set for each of those

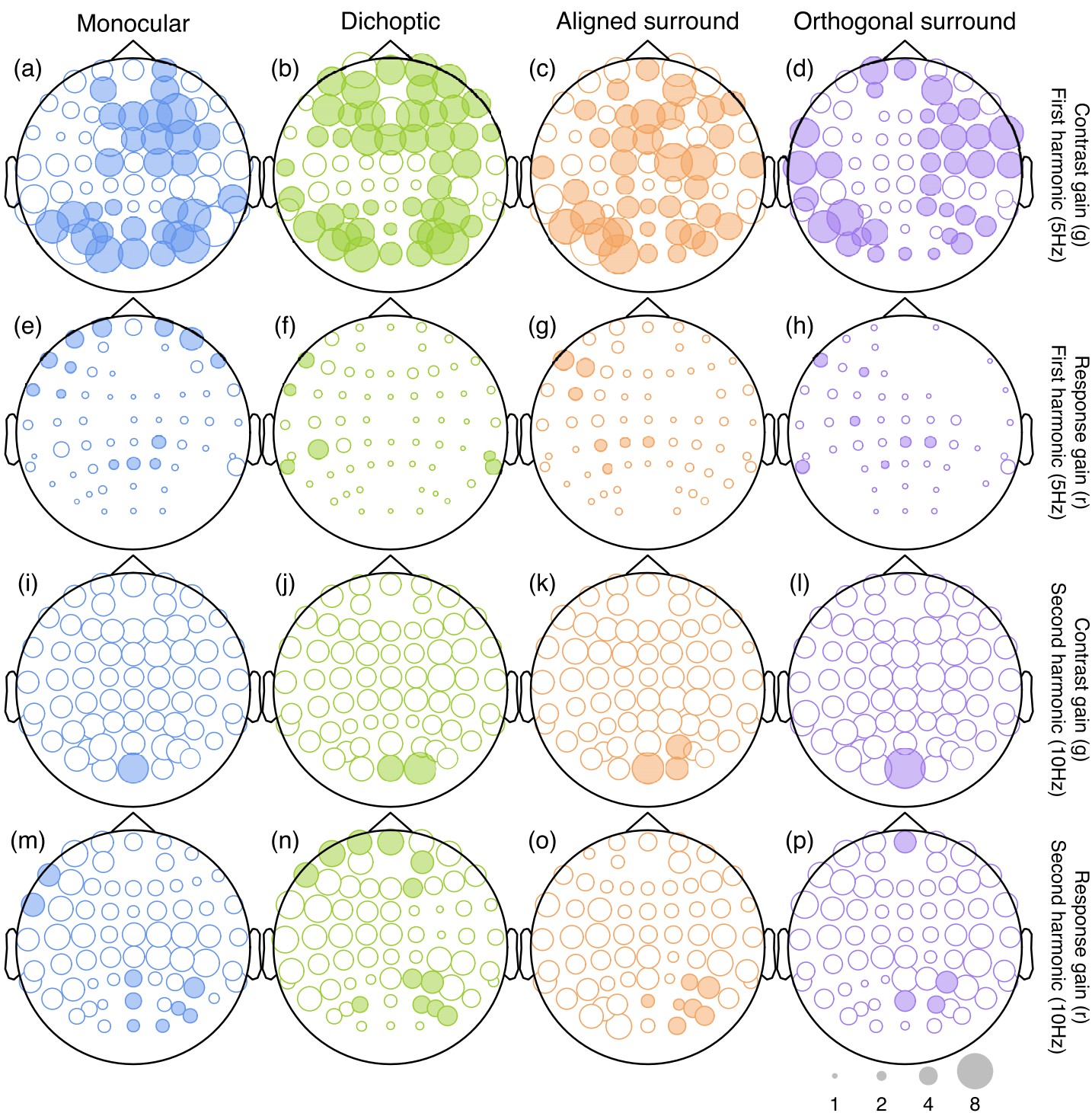

**Fig 5. Scalp plots summarising the suppressive weights for contrast and response gain from the Bayesian hierarchical model, fitted to data from the low mask contrast experiment.** Symbols are filled white when the 95 percent highest density interval of the posterior parameter distribution includes 1 (implying no credible contribution from that type of suppression), and shaded when it exceeds 1 (implying credible evidence for suppression). Larger symbols correspond to stronger suppression (see the scale in lower right corner), but parameters implying facilitation (values <1) are not plotted.

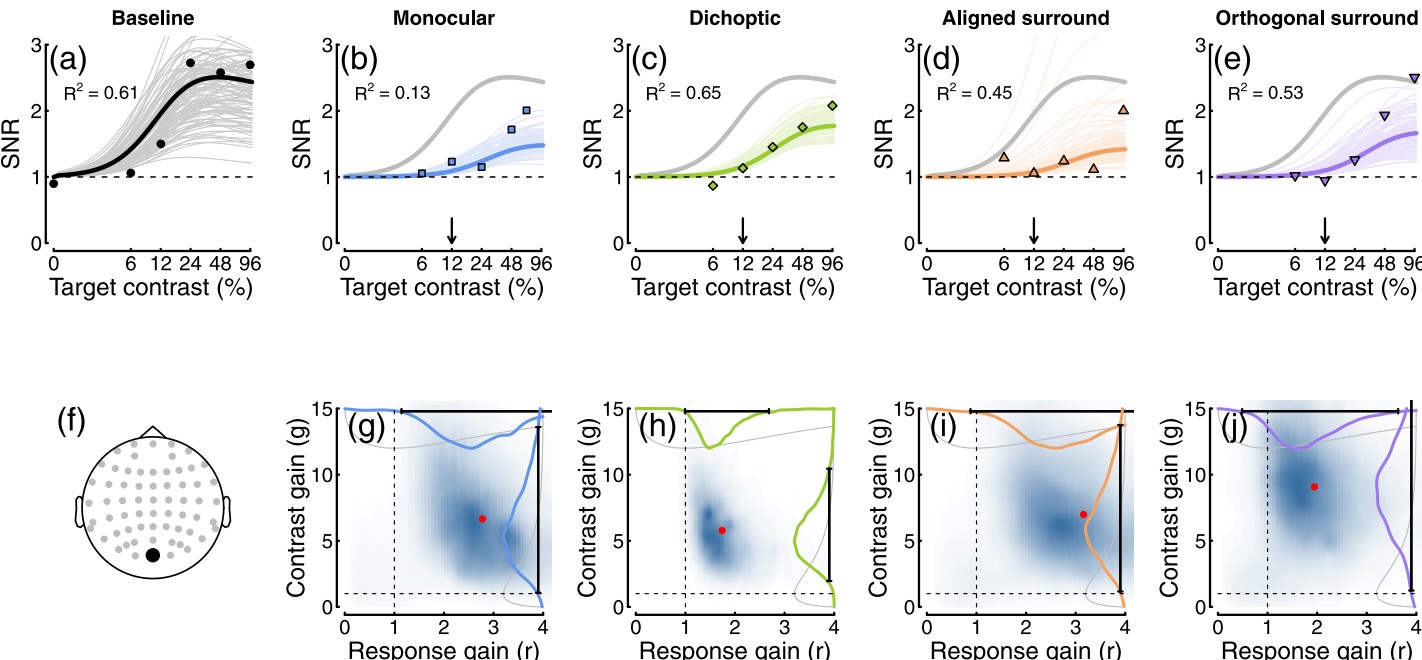

**Fig 6. Contrast response functions at the second harmonic frequency.** Plotting conventions mirror those in Fig 4. Note that the lower SNR at 10 Hz results in noisier data and less precise posterior estimates than at 5 Hz.

participants). The results for all four mask types are shown in Fig 8B–8E. At both the first and second harmonic frequencies, the target response decreased as a function of mask contrast. For the highest contrast monocular and dichoptic masks, this resulted in an almost complete suppression of the target response (SNR ∼ 1). For the surround masks at the first harmonic there was still a substantial signal even with the highest (96%) contrast masks.

We calculated a modified saturation index that takes into account the inversion of the functions. This was defined as the difference between responses at the highest two mask contrasts (48%–96%), scaled by the minimum of the function (we adjusted the index for the monocular mask to take into account the slightly lower mask contrast used to avoid clipping). Again, positive values imply acceleration, values of 0 saturation, and negative values supersaturation, but this time applied to the mask signal. At electrode *Oz*, the saturation index was near or below zero for monocular and dichoptic masks at the first harmonic, and surround masks at the second harmonic. Across the scalp (Fig 8F–8I), a range of saturation indices were apparent, though the mean index overall was positive (SI = 0.1).

## 4 Discussion

We asked whether suppression from four types of mask could be best explained by changes in contrast gain or response gain. A re-analysis of data from 16 studies was inconclusive, most likely due to methodological heterogeneity across studies. Data from two new SSVEP experiments showed that at the first harmonic frequency, all four mask types were best explained by contrast gain control, with minimal influence from response gain. However at the second harmonic frequency, both types of gain control were apparent. There was also evidence that the strength of suppression, particularly from the surround, increased away from the occipital pole. Finally, we asked whether suppressive signals saturate before impacting the target, and found some evidence of this for monocular and dichoptic masks at the first harmonic, and

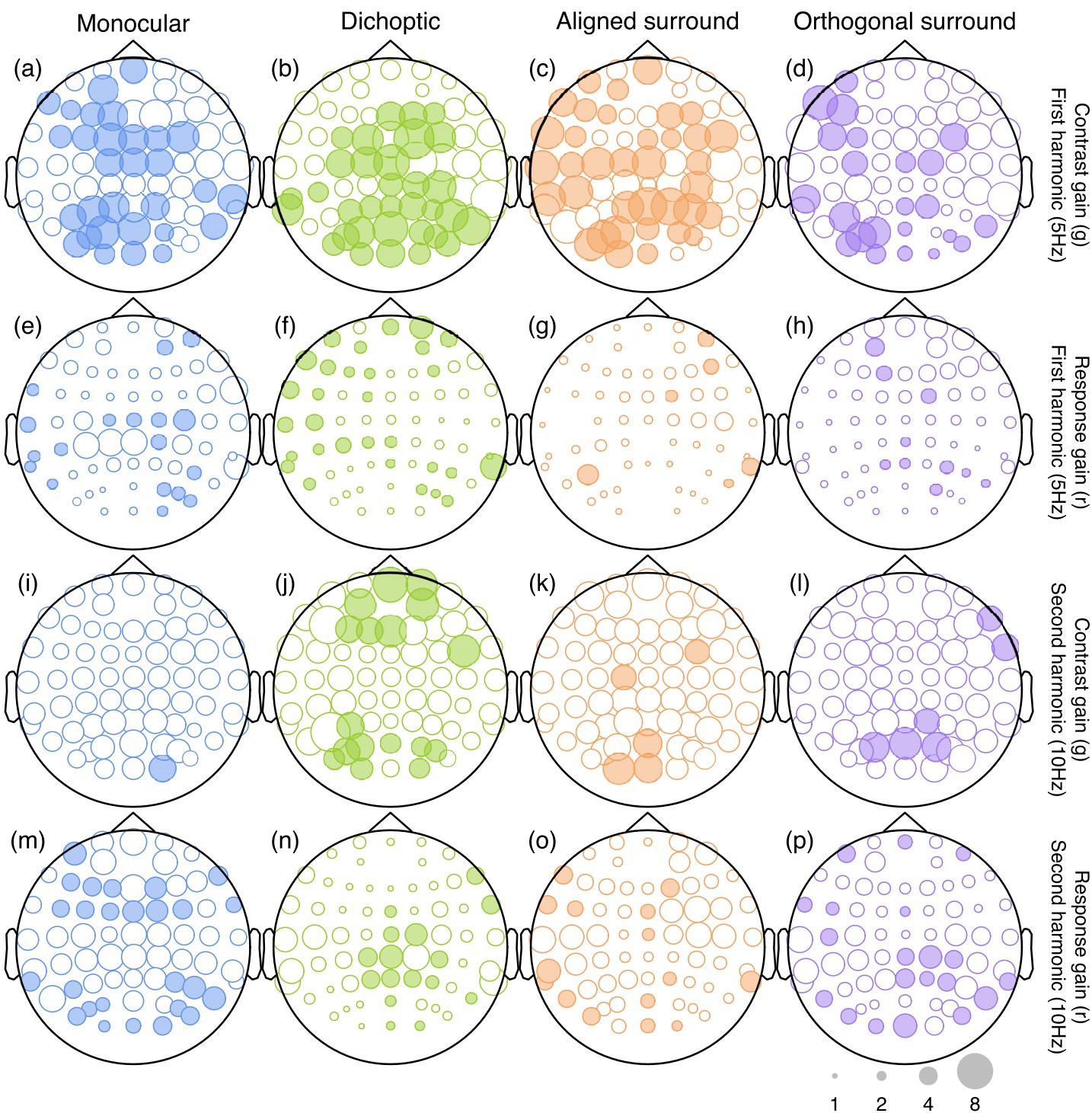

**Fig 7. Scalp plots summarising the suppressive weights for contrast and response gain from the Bayesian hierarchical model, fitted to data from the high mask contrast experiment.** Plotting conventions are as for Fig 5.

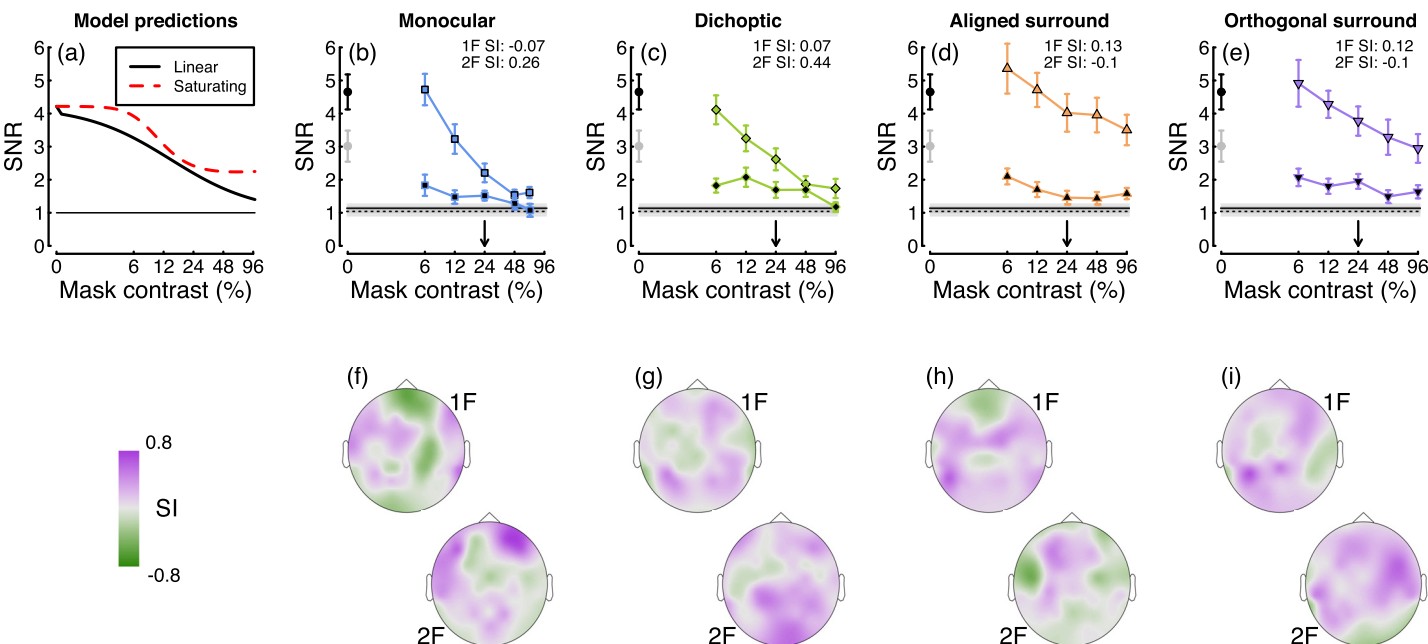

**Fig 8. Summary of the effects of varying mask contrast.** Panel A shows the predictions of a gain control model ([Eq 1]) for different levels of mask contrast. In the linear model (black), the suppressive signal is a linear function of mask contrast. In the nonlinear model (red), the suppressive signal has itself passed through a nonlinear transducer function before suppressing the target. Panels B-E show empirical data for four mask types, at the first and second harmonic frequencies (black borders and black fills, respectively). Error bars and shaded regions show ±1 standard error of the mean across N = 21 participants. Panels F-I show how the modified saturation index varies across the scalp.

surround masks at the second harmonic. We now discuss whether other experimental paradigms, such as animal electrophysiology, magnetic resonance imaging, and psychophysics, provide evidence for contrast or response gain, and consider different hypotheses about the purpose of suppression in the brain.

## 4.1 Contrast and response gain in other experimental paradigms

In single unit electrophysiology studies, overlay masking has long been attributed to contrast gain control, following the influential work of Heeger [2] (see also [44–46]). Although subsequent work has questioned whether this suppression arises cortically or subcortically [4, 47], the data remain consistent with a contrast gain effect. For dichoptic and surround masks, Sengpiel et al. [13] demonstrated that response gain provided a better explanation of responses in V1 neurons (see also [6, 48, 49]). However these effects were layer-dependent, with layer 4 showing response gain effects, and other layers more consistent with contrast gain. Additionally, Sengpiel and Blakemore [48] found strong response gain effects from the abrupt onset of a dichoptic mask, but only when the target was already present at mask onset. This suggests that interocular suppression may comprise multiple mechanisms, consistent with a variety of temporally-dependent perceptual suppression effects associated with binocular rivalry (e.g. [50]). Recently, it has also been demonstrated that suppression of stimulus-evoked gamma responses can be explained by contrast gain effects [51].

Some studies have used fMRI to investigate different types of suppression, though the analysis is less straightforward than with SSVEP as it is not possible to tag the target and mask at different frequencies to dissociate their effects. Moradi and Heeger [52] measured fMRI responses to gratings with monocular and dichoptic cross-oriented masks. Their results were

well-described by a contrast gain control model, though there are caveats in the interpretation given that the BOLD signal provides a single measure of the combined response to target and mask stimuli. Zenger-Landolt and Heeger [19] measured surround suppression by carefully locating voxels that responded only to the target location in an independent localiser experiment. Their surround suppression data appear consistent with a response gain change, though they did not fit a model to confirm this.

Psychophysical masking studies have traditionally assumed contrast gain effects from all mask types, following the seminal modelling work of Foley [53]. However, in principle elevation of detection thresholds can also be obtained through a response gain effect, as both manipulations reduce the signal-to-noise ratio and reduce sensitivity. The two effects are difficult to dissociate at detection threshold, but have differential effects above threshold, for example using contrast matching and discrimination paradigms. Some studies have considered model arrangements that are equivalent to response gain effects, notably in the context of surround masking [19, 54] and noise masking paradigms [55]. In surround masking experiments, facilitation effects are sometimes observed, particularly in matching experiments when the central target is of higher contrast than the surround [54, 56, 57], and these can be explained by an increase in response gain. We see some evidence of this in our SSVEP data, where surrounds enhance the response to the highest contrast targets (see Fig 4D). Another interesting result is that of Watanabe et al. [58], who found that during interocular suppression from binocular rivalry, contrast discrimination thresholds are increased. This result (subsequently replicated by [59, 60]) is consistent with a response gain effect, but not a contrast gain effect.

Overall, our SSVEP findings complement other work in the literature on understanding masking effects. Differences between our findings and results from other paradigms might be a consequence of SSVEP signals primarily indexing particular classes of neurons [61] or cortical layers [62]. In addition, threshold psychophysics is typically assumed to probe only the most sensitive mechanisms that respond to a target, whereas SSVEP measures the full population response, which may behave differently.

## 4.2 Model variations and other types of suppression

The model we present here is agnostic regarding the precise way in which contrast gain control is implemented, as we used only a single mask component at a fixed contrast in each experiment. Therefore any increase to the denominator term will have the required effect, and applying a modulatory weight ($g$) to the saturation constant ($Z$) is a parsimonious approach. However more realistic models typically include a distinct denominator term representing the mask contrast, sometimes with a scaling factor (weight). Early formulations of the contrast gain control model assumed that each suppressive signal on the denominator is raised to an exponent before impacting the target mechanism [53], and that the gain pool was untuned [2]. This implementation is sufficient for many data sets, however there is abundant evidence that suppression is tuned for both orientation and spatial frequency [3, 63]. Furthermore, psychophysical work comparing suppression from grating and plaid masks concluded that the mask components are summed linearly before impacting the target channel [64]. A similar arrangement also gives the best fit to multi-unit and gamma oscillation data recorded from cat visual cortex in response to plaids [51]. In principle any of these variant models would give an equally good fit to our data set, but the parameterisation we chose allowed for clean comparison with response gain effects.

More generally, there are other ways in which masks can affect target responses that we do not consider here. For example, subtractive inhibition is a plausible mechanism both in the retina [65] and cortex [66]. It has also been suggested that overlay masking is largely due to

saturation of LGN responses implemented as a subtractive 'push-pull' effect [47], though this manifests as a contrast gain effect in cortex, and besides cannot explain suppression from surround or dichoptic masks. In psychophysical studies, masks can affect target detectability through other mechanisms like uncertainty [67], distraction [68] and crowding [69] that may impact the decision process rather than the low-level sensory response that is measured using SSVEP. However we do not require any of these processes to explain our data, and so do not consider them further.

### 4.3 Why do SSVEP signals sometimes saturate?

Our results include examples of saturation at high target contrasts, particularly at the second harmonic frequency (see Fig 3C). Second harmonic responses have previously been associated with different classes of neuron from the fundamental response (e.g. [61]), and imply the presence of one or more nonlinearities, such as squaring or rectification [20, 42]. In principle a stronger nonlinearity might also result in the greater saturation we observe here at 10Hz. More generally, the studies summarised in Fig 2 show a wide range of behaviours at the first harmonic, from acceleration (see especially the Chadnova, Smith and Vanegas studies) through to strong supersaturation (see especially the Burr and Tsai studies). Saturation indices for these studies range from -0.44 to 0.49 (see Table 1).

There are several possible explanations for these differences. One possibility is that studies using a sweep-VEP paradigm, in which the stimulus contrast increases during a trial, might suffer from in-trial sequential adaptation effects that cause saturation towards high contrasts. Another explanation concerns the size and spectral content of the stimulus. Large stimuli will be subject to lateral suppression between adjacent areas of the stimulus, via the same mechanism that causes surround suppression. This would be expected to cause saturation at higher contrasts, where suppressive effects are strongest. The same argument holds in the Fourier domain for stimuli that are spectrally broadband, and so stimulate neurons responsive to a range of orientations and spatial frequencies. These neurons will mutually inhibit each other via the overlay masking pathway, again causing saturation. Several studies used large broadband noise textures as stimuli, most notably the study of Tsai et al. [15], which shows some of the strongest supersaturation. Other paradigms use small, spatially local patches of narrowband sine wave grating (e.g. [38]), which instead tend to produce accelerating responses. (Super)saturation may therefore provide an additional estimate of suppression strength, that could be leveraged in studies investigating group or individual differences in suppression. For discussion of the potential importance of supersaturation in individual neurons, see [70].

### 4.4 What is suppression for?

There have been many suggestions about the purpose of gain control suppression in the brain. These include reducing redundancy, sharpening tuning, and optimising sensitivity for the current environment (see [1], for further discussion). Recent work has demonstrated that suppression between neurons can be *reweighted* based on recent stimulation history [71]. Specifically, neurons that fire at the same time come to suppress each other more strongly [72]. This is a novel type of adaptation that is quite different from traditional paradigms in which a single stimulus is used as an adaptor. It suggests that gain control processes are dynamic rather than fixed, and can be modulated by past stimulation. Recently, we [23] pointed out that a gain control model of signal combination appears to be implementing statistically optimal combination of noisy signals, and suggested that suppression is the mechanism by which multiple cues are weighted. A prediction that follows from this idea is that the gain control process should be flexible enough to change the extent of suppression between two

signals to dynamically suppress noisier inputs. This prediction is consistent with the normalization reweighting idea, because when two stimuli are presented together, the covariance between the neurons they activate will increase, and they should suppress each other more. Normalization reweighting has also been demonstrated psychophysically for both overlaid and surround masks [73], suggesting that this process might operate across multiple suppressive pathways.

## 5 Conclusions

We asked if four types of masking are best explained by contrast gain or response gain effects when measured using steady-state EEG. A computational meta-analysis of 16 existing studies proved inconclusive, so we conducted two new experiments. The results show that overlay, dichoptic and surround masks are all best described by contrast gain effects for responses at the first harmonic. Suppression at the second harmonic involved a combination of contrast and response gain effects. We also found some evidence that suppressive signals saturate before impacting the target, though this was not consistent across mask type and response frequency. Although suppression from different mask types involves distinct anatomical pathways, gain control processes appear to serve a common purpose, which we suggest might be to suppress less reliable inputs.

## Author Contributions

**Conceptualization:** Daniel H. Baker, Greta Vilidaite, Alex R. Wade.

**Data curation:** Daniel H. Baker.

**Formal analysis:** Daniel H. Baker.

**Funding acquisition:** Daniel H. Baker.

**Investigation:** Daniel H. Baker, Greta Vilidaite, Alex R. Wade.

**Methodology:** Daniel H. Baker, Greta Vilidaite, Alex R. Wade.

**Project administration:** Daniel H. Baker.

**Resources:** Daniel H. Baker, Alex R. Wade.

**Software:** Daniel H. Baker.

**Supervision:** Daniel H. Baker, Alex R. Wade.

**Visualization:** Daniel H. Baker.

**Writing – original draft:** Daniel H. Baker.

**Writing – review & editing:** Greta Vilidaite, Alex R. Wade.

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
