## [Decision Letter · Decision Letter 0]

30 Aug 2021

Dear Dr Baker,

Thank you very much for submitting your manuscript "Electrophysiological measures of visual suppression" for consideration at PLOS Computational Biology. As with all papers reviewed by the journal, your manuscript was reviewed by members of the editorial board and by two independent reviewers. The reviewers appreciated the attention to an important topic. Based on the reviews, we are likely to accept this manuscript for publication, providing that you modify the manuscript according to the review recommendations.

Sincerely,

Tianming Yang

Associate Editor

PLOS Computational Biology

Wolfgang Einhäuser

Deputy Editor

PLOS Computational Biology

[LINK]

Reviewer's Responses to Questions

**Comments to the Authors:**

Reviewer #1: This manuscript has strengths and weaknesses. Overall the strengths outweigh the weaknesses and with appropriate revision, it is likely that this manuscript will make a significant contribution to our understanding of contrast masking.

Review of previous contrast masking studies

Strength: The manuscript evaluates 16 previous studies that examined how a mask interferes with the visibility of a target and concludes that these studies taken together do not show clear evidence for either contrast gain (where the mask reduces the sensitivity to the target) or response gain (where the mask reduces the maximum response evoked by the target). From their analysis of these studies, which include masks that are superimposed or surround the target, or are presented in the other eye (dichoptic), they conclude that there is no clear evidence for either contrast or response gain.

Weakness: 1. The review seems to combine examples that show different changes to the contrast response function in the presence of a mask (e.g., it appears that the 3.3 Hz and 5.5 Hz data from Candy et al, 2001 that show facilitative and suppressive trends in the presence of a superimposed orthogonal mask are averaged). Also, the review does not seem to take into account data from extra-striate cortex for dichoptic masks (from Hou et al, 2020). More information about the exact data sets used from these studies could be listed in Table 1.

Furthermore, a consideration of the patterns of responses across the cortical surface in these studies, if available, so that they can be compared to the measurements across electrodes made in their study.

Two-stage model

They use a two-stage model where the first stage fits a canonical gain control model to the unmasked data. They use a version of the model with 3 parameters to describe the contrast response function: a term that determines sensitivity (Z), another term that determines maximum response (Rmax) and a term (p) that determines whether the function accelerates or saturates. To incorporate the effect of the mask, they use a second-stage that modifies sensitivity and response gain. This model is used to analyze data from previous studies as well as their own data where they examine the effect of overlay, surround and dichoptic masks on the same targets.

Strengths: The modelling produces elegant summaries of previous data, despite the caveats noted above

Weakness: The model only considers the effect of the mask on contrast and response gain. However, their steady-state EEG data (first harmonic) show that the mask appears to have a significant effect on the acceleration parameter p, where the contrast response function goes from saturating without the mask to either accelerating (Fig 4b) or super-saturating (Fig 4 c, d, e) in the presence of the mask. Clearly the changes in saturation detract from a simple narrative centered on contrast and response gain, but it appears that this parameter is needed to better characterize the effect of the mask on the contrast response function.

The estimated model fits to the second harmonic term of the evoked response in the presence of a mask are particularly poor. Granted, the SNR of the second harmonic term is lower and the data are therefore noisier. It is not clear that the inclusion of the p term (acceleration) would improve the fit of the model, but a justification for not including the p term would be helpful.

In general, it would be interesting to determine if including the change in the p parameter due to the mask, will change the reported effect of the contributions of contrast and response gains to the previous studies, and to their own first harmonic and second harmonic data

Minor point: Line 246. The authors mention that overall the first harmonic responses accelerate (median SI value 0.10). However, the preceding text says the SI values corresponding to acceleration are greater than 1

Reviewer #2: The authors characterized suppressive effects in SSVEP signals under conditions with different stimulus configurations. They concluded that cross-orientation suppression, interocular suppression and surround suppression can all be characterized in the form of contrast gain control for the first harmonics of SSVEP, but the second harmonics showed both contrast and response gain effects. The authors also explore changes across different electrode sites and different response frequencies and found that suppression generally became stronger at more lateral electrode sites, which suggests that suppression builds up across successive stages of processing. There have been many electrophysiological studies on the normalization properties in animal researches and human EEG. Although animal studies have reached conclusion on gain control effects for different stimulus properties, studies on EEG signals from human brain are not conclusive, as the authors pointed out in their meta-analysis on data from 16 published studies. The results based on newly collected data provided a more systematical pictures about gain control effects in SSVEP. The findings in this study is interesting and the paper is easy to follow. I generally have no serious concern with the work, but there are still a few comments or questions that the authors should address.

1) The current title ‘Electrophysiological measures of visual suppression’ is a little overstated. ‘Electrophysiological’ includes studies with intracellular, extracellular recordings and EEG recordings. A more specific title might be better.

2) Normalization Model and its fitting: a) there is no information about how well normalization model explains the meta-data and the data collected by new experiment. The authors should add this piece of information (the performance of the model on explaining the data) in the study (maybe in the paragraph between line 249 and 266). b) Many forms for gain control effects has been used, but only one specific form of gain control model was chosen in this study. Among different models, is the chosen one the best to explain the data? I understand that current study is modeling SSVEP signals, which might have a model different from models for spike activity, but at least there should be some discussions about why the chosen is good compared to other models for population response (VEP, the LFP and the MUA). The models for the normalization of the LFP in visual cortex in a recent paper by Wang et al. (2021, Superimposed gratings induce diverse response patterns of gamma oscillations in primary visual cortex) might be worth considering and discussing. c) It is clear for the procedures of model fitting with two steps, but I could find the reason why the authors fitting the data in this way. A few sentences to justify such fitting procedures might be good for readers. Is the two-step fitting similar to fitting a complete model to the data with one step?

3) It is interesting to see different gain control effects between first and second harmonics: what is the cause for the difference? Some discussion about the difference or the origins of F1 and F2 will be good. Is there any possibility that the second harmonics is contaminated/affected by alpha rhythm around 10 Hz? In figure 3a, there seem to be dips around the two harmonic frequencies (5Hz and 10Hz), is this due to spectral analysis or normalization for power spectrum?

4) Lines 186-187, ‘each condition were then coherently averaged (i.e. taking both the phase and amplitude into account)’. It is not clear what coherently averaged mean. Please clarify this procedure.

5) Lines 220-221, ‘However, the 95% highest density interval of the group posterior distribution for contrast gain control, shown in the final row, overlapped with…’ This is not a complete sentence.

6) Lines 280-282: There is no statistics report for the comparison between surround suppression and monocular/dichoptic suppression, and in line 292-285, the comparison between parietal regions and occipital regions.

7) Discussion ‘Contrast and response gain in other experimental paradigms’, Not all normalization can be characterized by contrast or response gain controls (see Wang et al.2021 mentioned in comment 2C)..

**Have the authors made all data and (if applicable) computational code underlying the findings in their manuscript fully available?**

Reviewer #1: Yes

Reviewer #2: Yes

PLOS authors have the option to publish the peer review history of their article (what does this mean?). If published, this will include your full peer review and any attached files.

Reviewer #1: No

Reviewer #2: No

Figure Files:

Data Requirements:

Reproducibility:

References:

---

## [Editor Report · Decision Letter 1]

30 Sep 2021

Dear Dr Baker,

We are pleased to inform you that your manuscript 'Steady-state measures of visual suppression' has been provisionally accepted for publication in PLOS Computational Biology.

Best regards,

Tianming Yang

Associate Editor

PLOS Computational Biology

Wolfgang Einhäuser

Deputy Editor

PLOS Computational Biology

---

## [Editor Report · Acceptance letter]

7 Oct 2021

PCOMPBIOL-D-21-01424R1 

Steady-state measures of visual suppression

Dear Dr Baker,

I am pleased to inform you that your manuscript has been formally accepted for publication in PLOS Computational Biology. Your manuscript is now with our production department and you will be notified of the publication date in due course.

With kind regards,

Zsofia Freund
